# Degradation Mechanism of Aflatoxin M_1_ by Recombinant Catalase from *Bacillus pumilus*E-1-1-1: Food Applications in Milk and Beer

**DOI:** 10.3390/foods13060888

**Published:** 2024-03-15

**Authors:** Xiaoyu Liu, Fangkun Zhao, Xianghong Wang, Kaige Peng, Chunyu Kang, Yaxin Sang

**Affiliations:** College of Food Science and Technology, Hebei Agricultural University, 289 Lingyusi Road, Baoding 071001, China; liuxiaoyu202202@126.com (X.L.); zhaofkk@163.com (F.Z.); wangxianghong73@sina.com (X.W.); pkgdmn@163.com (K.P.); kangc-y@126.com (C.K.)

**Keywords:** catalase, biological degradation, catalase-mediator system, degradation products, hepatotoxicity

## Abstract

A bacteria capable of degrading aflatoxin M_1_ (AFM_1_) was isolated from African elephant manure. It was identified as *Bacillus pumilus* by 16s rDNA sequencing and named *B. pumilus*E-1-1-1. Compared with physical and chemical methods, biological methods have attracted much attention due to their advantages, such as thorough detoxification, high specificity, and environmental friendliness. This work aimed to study the effects of a recombinant catalase (rCAT) from *B. pumilus*E-1-1-1 on the degradation of AFM_1_ in pattern solution. The degradation mechanism was further explored and applied to milk and beer. Kinetic Momentum and Virtual Machine Maximum values for rCAT toward AFM_1_ were 4.1 μg/mL and 2.5 μg/mL/min, respectively. The rCAT-mediated AFM_1_ degradation product was identified as C_15_H_14_O_3_. Molecular docking simulations suggested that hydrogen and pi bonds played major roles in the steadiness of AFM_1_–rCAT. In other work, compared with identical density of AFM_1_, survival rates of Hep-G2 cells incubated with catalase-produced AFM_1_ degradation products increased by about 3 times. In addition, degradation rates in lager beer and milk were 31.3% and 47.2%, respectively. Therefore, CAT may be a prospective substitute to decrease AFM_1_ contamination in pattern solution, milk, and beer, thereby minimizing its influence on human health.

## 1. Introduction

Aflatoxin is a naturally occurring toxic fungal metabolite, primarily generated by *Aspergillus flavus* and *Aspergillus parasiticus*, with high carcinogenicity and mutagenicity effects [1]. There are 20 types of aflatoxins, including aflatoxin B_1_ (AFB_1_), aflatoxin B_2_ (AFB_2_), aflatoxin G_1_ (AFG_1_), aflatoxin G_2_ (AFG_2_), aflatoxin M_1_ (AFM_1_), and aflatoxin M_2_ (AFM_2_) [2]. AFB_1_ is the most toxic and carcinogenic. AFM_1_ is a hydroxyl metabolite of AFB_1_ with similar poisonousness effects and often occurs in milk, and consequently in other dairy products [3]. Studying AFM_1_ in food is significant as it may pose special risks to children. In recent years, breweries have been developed and beer consumption in China has increased. However, contamination with mycotoxins in beer raw materials has been the source of beer safety problems [4]. Since aflatoxin cannot be completely removed at present, it can only be controlled within a relatively safe dosage range to ensure people’s health. Different nations have established maximum residue limits for AFM_1_; for example, 0.5 µg/L in China and USA and 0.05 µg/L in the European Union [5,6].

Currently, AFM_1_ can be removed by physical, chemical, and biological methods [7]. However, physical and chemical degradation methods have limitations, including high costs, chemical residues, and an impact on food nutrients, rendering them unsuitable in practice. The biodegradation of AFM_1_ has the advantages of high specificity, low pollution, and mildness to nutrients [8]. Therefore, the biodegradation of AFM_1_ has become a research hotspot in recent years. There are two pathways for the biodegradation of AFM_1_, bacterial adsorption and enzymatic degradation. Bacterial adsorption is incapable of degrading toxins and only transfers them, thereby requiring enzymatic degradation after adsorption. However, enzymatic degradation directly converts toxins into non-toxic substances [9].

Catalase (CAT) is one of the key enzymes of the biological defense system, acting as an enzymatic scavenger in all living organisms to eliminate oxygen radicals [10]. CAT catalyzes the decomposition of H_2_O_2_ to H_2_O and O_2_ in both eukaryotes and prokaryotes. Due to its excellent catalytic efficiency, CAT has been used in a wide range of textile and biomedical applications, especially as a food additive for cheese, milk, and dairy products in food industries [7].

Currently, studies on the biodegradation of aflatoxins mostly focus on the identification of substances with aflatoxin degradation abilities, while relatively few studies have focused on their specific structures, degradation mechanisms, and products. In a previous study, Qi et al. reported that pulse light could degrade aflatoxin, with an analysis of the degradation products performed by the authors [11]. Nonetheless, cytotoxicity studies are needed to verify specific changes in toxicity. Li et al. reported that *Tetragenococcus halophilus* could degrade aflatoxin B_1,_ with the degradation products identified as C_17_H_10_O_7_ and C_16_H_12_O_5_ [12]. In addition to the identifying the necessary mechanisms in aflatoxin degradation, CAT extraction from mammalian liver or plant tissue is time-consuming and costly in terms of production and applications. Microorganisms are the preferred sources of research and application of catalytic enzymes due to their short growth cycles, low cultivation costs, and ease of operation [7]. In this study, we recombinantly expressed a CAT enzyme with aflatoxin degradation abilities from *B. pumilus*E-1-1-1, and investigated its degradation and cytotoxicity mechanisms after degradation. The recombinant enzyme was further researched for its capacity to degrade AFM_1_ in different foods.

## 2. Materials and Methods

### 2.1. Strains, Plasmids, and Chemicals

A bacterium capable of degrading aflatoxin M_1_ (AFM_1_) was isolated from African elephant manure using coumarin and AFM_1_ as screening indicators, respectively. It was identified as *B. pumilus* by 16s rDNA sequencing and named *B. pumilus*E-1-1-1. The strain was stored in 30% glycerol at −80 °C. Plasmid extraction was performed using a Tiangen p (Beijing, China) lasmid extraction kit according to manufacturer’s instructions.

### 2.2. PCR Amplification of the CAT Gene and Cloning into an Expression Vector

CATF and CATR (forward and reverse primers) were designed and analyzed by Primer-BLAST (Table 1). The whole *B. pumilus*E-1-1-1 genome was extracted using a bacterial genomic DNA extraction kit (Trans Gen Biotech Co., Beijing, China) and used as a template for the amplification of target genes. PCR reactions were performed for 30 amplification cycles, which involved initial denaturation at 98 °C for 30 s, annealing at 45 °C for 30 s, an extended step at 72 °C for 1min, and a final extension at 72 °C at 10 min. Amplicon products of approximately 1600 bp were obtained, analyzed on a 1.5% agarose gel, and observed using a gel imaging system.

The expression vector for cloning PCR amplification products was digested with two restriction enzymes, *Bam*HⅠ and *Not* Ⅰ, to produce compatible sticky ends for restriction ligation cloning to express a 75 kDa protein (Figure 1). The purified vector and *CAT* gene were ligated with T4 DNA ligase (Trans Gen Biotech Co., Beijing, China) at 16 °C overnight. After cloning the *CAT* gene into the PET28a vector, the vector was transformed into *Escherichia coli* (*E. coli*) BL21(DE3) competent cells by heat shock method. Clones containing the *CAT* gene were confirmed by PCR using vector-specific primers (T7 and T7-term) and gene-specific primers. Extracted plasmids and primers were sent to Genewiz Company for sequencing verification.

Recombinant catalase (rCAT) was then expressed in *E. coli* BL21(DE3) cells at 30 °C. To induce protein expression, the concentration of isopropyl β-D-1-thiogalactopyranoside (IPTG) was set to 0.5 mmol/L. After expression induction, the precipitate was collected by centrifugation (6500 r/min, 10 min) and redissolved in phosphate buffer. Protein expression levels were detected by SDS-PAGE and then proteins were purified by Ni^2+^-NTA columns using His-tag purification resin (Solarbio Life Sciences, Beijing, China).

### 2.3. Quantitative Analysis of rCAT

RCAT protein concentrations were measured by BCA protein assay kit, and its activity was determined by the hydrogen peroxide method [13]. Specific steps are enumerated in Appendix A. All trials were conducted three times with purified enzyme. One unit of rCAT oxidated 1 μmol of H_2_O_2_ per minute.

### 2.4. Measurement of AFM_1_ Concentrations by HPLC

The concentration of AFM_1_ was measured by HPLC, with steps enumerated in Appendix A.

### 2.5. AFM_1_ Degradation by rCAT

The starting concentration of AFM_1_ standard solution stored at −20 °C was 1 mg/mL. This standard solution was diluted with acetonitrile into working solutions of different concentrations. The AFM_1_ working solution (2 μg/mL) was fully reacted with rCAT (1 U/mg) at 40 °C for 72 h. The reaction was terminated by adding 5 mL water: methanol (30:70, *v/v*) to reaction systems, while 5 mL of chloroform was added to extract AFM_1_. The mixed solution was ultrasonicated and shaken for 10 min and then separated through a separatory funnel; the lower solution was collected in a 100 mL cocktail flask. The whole operation was repeated three times, and the reaction system was vapored to dryness and then reconstituted twice with 500 mL acetonitrile. The AFM_1_ obtained by reconstitution was passed through a filter membrane and injected into a HPLC system for detection and quantification analyses. The remaining AFM_1_ in the pattern solution was collected and analyzed according to the following equation: AFM_1_ degradation (%) = (AC_initial_ − AC_final_)/AC_initial_ × 100

As shown in the formula, AC_initial_ denotes the starting concentration of AFM_1_, AC_final_ denotes the final concentration of AFM_1_, and the AFM_1_ concentration was obtained from the AFM_1_ standard curve. The pH was evaluated in the 4–10 range (4, 5, 6, 7, 8, 9, and 10) to determine the optimal pH. Then, the reaction temperature was evaluated from 20 to 70 °C (20, 30, 40, 50, 60, and 70 °C). Different metal ions in AFM_1_ degradation studies were also analyzed to determine which ones best promoted reactions.

### 2.6. Degradation of AFM_1_ Times

The pattern solution contained rCAT, AFM_1_, and phosphate buffer (pH 7.4, 75 mmol/L). Samples were obtained at different times (0, 1, 2, 6, 12, 24, 48, and 72 h), and residual AFM_1_ was collected and quantified, which in turn was used to calculate the degradation rate of rCAT. All experiments were carried out in triplicate.

### 2.7. Measurement of Kinetic Parameters

Kinetic Momentum (K_m_) and Virtual Machine Maximum values (V_max_) for rCAT (1 U/mL) were measured at 24 h under optimum reaction conditions for enzyme activity. The concentrations of AFM_1_ were 0.01, 0.05, 0.1, 0.15, 0.2, 0.25, 0.3, 0.4, 0.5, and 0.6 μg/mL. Trials were conducted in triplicate and compared with the control group.

### 2.8. Liquid Chromatography–Mass Spectrometry (LC–MS) Analysis for the Identification of Degraded Products

rCAT-mediated AFM_1_ degradation products were analyzed by UPLC (ACQUITY UPLC System, Agilent Corporation(Santa Clara, CA, USA) together with a quadrupole orthogonal acceleration time-of-flight mass spectrometer (TOF-MS, LCT Premier, Waters Corporation(Milford, MA, USA). ESI-MS tests were conducted in positive ion mode. The mobile phase was eluted at a velocity of 0.3 mL/min and contained 0.1% formic acid in water and acetonitrile. The following gradient was adopted: the acetonitrile concentration increased from 10% to 90% within 1–12 min and from 90% to 10% acetonitrile within 12–15 min. A Waters UPLC BEH C18 column (1.7 μm, 2.1 × 100 mm) was operated for split with a 5 μL sample injection volume.

The m/z value was evaluated based on papers representing the degradation products of AFB_1_ and AFM_1_ through different pathways. In addition, determined structures were arranged to obtain assumed degradation pathways.

### 2.9. Cell Viability Assay

Human hepatocarcinoma cells (Hep-G2) were selected as a cell model as the liver is the main target organ for aflatoxin [14]. Cells (Pricella CL-0103; Incubation to logarithmic stage) were continuously cultivated in Dulbecco’s Modified Eagle’s Medium (DMEM) complete medium (10% *v/v* fetal bovine serum, 100 units/mL penicillin, and 100 μg/mL streptomycin) and cultured at 37 °C in 5% CO_2_ until cell adhesion. Then, a total of 5 × 10^4^ cells were seeded in 96-well plates and cultured overnight [15].

Cells were divided into different groups by AFM_1_ concentration (0, 50, 100, 150, 200 ng/mL, and AFM_1_ degradation products) and then cultured for 48 h. Cell viability was determined using CCK-8 assays (Beyotime Institute of Biotechnology, Suzhou, China) where 10 μL CCK-8 reagent solution was added to each well, and reaction mixtures cultured for 2 h at 37 °C.

After incubation, the absorbance of each well was determined at 450 nm using an ELISA reader (Thermo Lab Systems, Milford, MA, USA) and percentage viabilities were estimated.

### 2.10. Homology Modeling and Molecular Docking

Docking simulations were conducted with MOE Dock in MOE v2014.0901 to explore combination modes between AFM_1_ and CAT. According to the X-ray crystal structure of catalase DR1998 from *Deinococcus radiodurans* (PDB entry 4cab.1A), which is the most homologous structure to *B. pumilus*E-1-1-1 CAT at 70.9%, the 3D structural model of *B. pumilus*E-1-1-1 CAT was structured using a homology modeling tool, the SWISS-MODEL (https://swissmodel.expasy.org/interactive, accessed on 8 September 2023) server. The 2D structure of AFM_1_ was obtained in ChemBioDraw 2014 and transformed to 3D in MOE v2014.0901 using energy minimization.

### 2.11. AFM_1_ Degradation in Beer and Milk

The existence of AFM_1_ in beer and milk, for removal by rCAT, was confirmed. An original solution of 2 μg/mL was obtained by diluting an AFM_1_ standard solution with acetonitrile. A working solution (100 μL of 200 ng/mL AFM_1_) was then added to reaction vessels, and liquid evaporation with nitrogen was performed. Subsequently, rCAT (1 U/mL) was added to ultrasonic degassed beer, and the optimum factors identified above were adopted. Residual AFM_1_ was collected and fractionated by HPLC (the method is the same as in Section 2.5).

### 2.12. Statistical Analysis

Statistical Package for Social Sciences (SPSS 22.0, Chicago, IL, USA) was used to perform analysis of variance (ANOVA) and Tukey’s test at 95% confidence levels (*p* < 0.05).

## 3. Results and Discussion

### 3.1. Purified Proteins, Concentrations, and Enzyme Activity of CAT

The full-length coding sequence of CAT is 1590 bp (Figure 2a) and codes for a 563 amino acid protein with a theoretical molecular mass of 75 kDa (Figure 2b). The CAT protein was consistent with the theoretically calculated molecular weight and appeared as a clear band on SDS-PAGE in Figure 2b. Xu et al. found that a catalase from *Pseudomonas aeruginosa* could degrade AFB_1_ of 55.6 kDa by 38.8%, which was the first time that the degradation of AFB_1_ by a recombinant catalase was reported [16]. AFM_1_ is a metabolite of AFB_1_ and has a very similar structure, differing by only one hydroxyl group [17]. The concentration of rCAT measured with the BCA protein concentration assay kit was 619.07 μg/mL. The activity of rCAT measured by the H_2_O_2_ method was 445.22 U. Based on the above definition of enzyme activity, the protein required for the degradation reaction was 1 U. Shi et al. found that a recombinant human catalase peaked at 600 U/mg at 60 h [16]. Nagy et al. successfully expressed a catalase from *Mycobacterium tuberculosis* and its activity was measured at 431 U/mg [17].

### 3.2. Characterization of AFM_1_ Degradation by rCAT in Pattern Solution

#### 3.2.1. pH

As shown in Figure 3a, the degradation rate of AFM_1_ had an initial up and then down trend in the pH 4–10 range, and reached a maximum at pH 7. Our pH optimization results showed that the enzyme had the highest activity in a neutral environment, with the highest AFM_1_ degradation rate exceeding 60%. However, it had lower activity in acid and alkaline environments. The AFM_1_ degradation rate decreased with increasing acid–base strength, with the lowest degradation of about 10%, suggesting that rCAT was a neutral enzyme. The above results showed that pH was one of the factors affecting the effectiveness of rCAT in degrading AFM_1_, and the best rCAT degradation effects were achieved at pH 7.

A previous study reported that a highly heat-resistant and efficient recombinant manganese-catalase from *Geobacillus thermopakistaniensis* exhibited its highest activity at pH 10 [18]. A novel recombinant manganese-catalase from *Bacillus subtilis* R5, with an optimal pH of 8, was the first cloned and characterized manganese-catalase from the *Bacillus* genus [19]. Zhang et al. reported that the optimum pH of a recombinant wheat catalase, with an ability to improve wheat-flour-processing quality, was 7.5 [20]. These findings generally agreed with the optimal pH (7.0) of recombinant CAT evaluated in this study.

#### 3.2.2. Temperature

The AFM_1_ degradation rate tended to increase and then decrease between 20 and 70 °C, reaching a maximum of 60% at 40 °C (Figure 3b). The lower AFM_1_ degradation rate might be due to incomplete enzyme activation when the temperature was less than 40 °C. When the temperature exceeded 40 °C, the degradation rate had a decreasing trend with increasing temperature, which could be the result of enzyme inactivation at progressively higher temperatures.

It has been reported that a rCAT from *Corynebacterium glutamicum* exhibited its highest activity at 30 °C and represented good thermal stability from 25 °C to 50 °C [21]. Philibert et al. reported that a new recombinant heme-catalase from *B. subtilis* 168, with an optimum temperature of 40 °C, improved oxidative stress in *B. subtilis* [22]. A previous study found a monofunctional rCAT with the optimal temperature of 4 °C from *Pimentiphaga sp*. DL-8 increased the specific activity of hydrogen peroxide by 9 times when compared to the parent strain [23]. In summarizing and comparing previous literature results, it was found that the optimal reaction temperatures of different rCATs varied considerably, probably due to different sources of recombinant enzymes. RCATs from the same genus (*B. subtilis*) had an identical optimal reaction temperature of 40 °C [22].

#### 3.2.3. The Effects of Metal Ion Addition

The effects of metal ions on enzymes refers to the impact of metal ion binding to enzymes during catalytic reactions, which generally includes activators and inhibitors. Figure 3c shows the influence of different metal ions on the removal of AFM_1_ by rCAT in pattern solution. The control group did not include metal ions, and they had a degradation rate of 62.2%. Compared to the control, among all metal ions, K^+^ and Na^+^ enhanced AFM_1_ degradation effects, while the other metal ions were inhibitory to degradation reactions. These results showed that in the presence of K^+^ and Na^+^, AFM_1_ degradation rates increased by 4.6% and 6.1%, respectively, compared to the control. On the contrary, Mg^2+^, Fe^2+^, Li^+^, and Ca^2+^ inhibited AFM_1_ degradation compared with the CK group. In particular, Li^+^ had the maximum inhibitory effects on degradation, with a degradation amplitude of 59.9%.

### 3.3. Degradation Times of AFM_1_ by rCAT

Figure 3d shows AFM_1_ degradation rates by rCAT within 72 h. The degradation rates of whole reactions rose sharply from 0 to 12 h, then levels attained a near-equilibrium state. The maximum degradation rate was 63.2% throughout the entire process. Therefore, the optimum cultivation time for rCAT to degrade AFM_1_ in pattern solution was 12 h.

### 3.4. Determination of Kinetic Parameters

Km and Vmax values were measured from equations in the bifold plot of AFM_1_ degradation in the presence of rCAT. Km and Vmax results for the rCAT degradation of AFM_1_ were 4.1 μg/mL/min and 2.5 μg/mL/min, respectively (Appendix A).

In the literature, the Km results of a commercial product for the degradation of mycotoxins (OTA and ZEN) were 20 and 10,710 μg/mL, and Vmax results were 0.068 and 23 ng/mL/min, respectively [24]. By comparison, the low Km and high Vmax values found in this study indicated that the recombinant enzyme produced by *B. pumilus*E-1-1-1 had high substrate affinity and catalytic efficiency, which would be advantageous for industrial applications.

### 3.5. Identification of Degraded Products by LC–MS

LC–MS analysis was conducted to further study the degradation products of AFM_1_ [25]. A total ion flow diagram in positive ion mode is shown in Appendix A, where only one obvious chromatographic peak was observed. The mass spectrum of this peak (Appendix A) showed that the 329.21 AU×min ion had the highest abundance, as the theoretical value of the AFM_1_ molecular weight was 328.27 AU*min. Therefore, the highest ion peak was [AFM_1_ + H]^+^, while 351.05 was [AFM_1_ + Na]^+^. These results suggested that this method could be used for subsequent degradation product analysis. 

A total ion flow diagram of the degradation product produced by rCAT in positive ion mode is shown in Appendix A, with three distinct chromatographic peaks. The product peak at the retention time of 8.73 min (Appendix A) was identified by MS analysis in Appendix A. It was obvious that Appendix A had four ion peaks, with m/z values of 111.0920, 173.0801, 242.2131, and 362.3284, respectively. The degradation product of AFM_1_ was C_15_H_14_O_3_ ([M + H 242.2131). By analyzing fragment ions and mass spectrometry information combined with qualitative software, a possible structure was deduced and showed that lactone and difuran rings of AFM_1_ were disrupted. The results also suggested that AFM_1_ product toxicity was greatly reduced during degrading processes (Figure 4). Previous studies on toxicity have shown that AFM_1_ has a cyclopentenone ring and a C8–C9 double bond that forms vinyl ether at the end of the furan ring. The furan ring of AFM_1_ is a toxic and carcinogenic group, and the double bond in its ultimate furan ring is the main factor causing toxicity [26].

Nikmaram et al. observed three main degradation products of AFM_1_; C_15_H_11_O_7_, C_17_H_15_O_9_, and C_15_H_13_O_7_, which were formed through high voltage atmospheric cold plasma (HVACP) treatment [27]. The authors also found that these three degradation products indicated that additive reactions happened on the double bond in the end of furan ring, consistent with degradation mechanisms in this paper. It was also shown that the double bond of the furan ring was oxidated in degradation processes [26].

### 3.6. Toxicity Analysis of CAT-Mediated AFM_1_ Oxidation Products

Hep-G2 cells were chosen as they are completely differentiated initial hepatocytes. The toxicity of AFM_1_ and its degradation products were assessed using CCK8 assays. Hep-G2 cells were supplemented with different concentrations of AFM_1_ as shown in Figure 5a. The results showed that with increasing concentration of AFM_1_, percentage viabilities in Hep-G2 cells were significantly lowered. However, in Figure 5a, Hep-G2 percentage survival rates were less than 50% when the AFM_1_ concentration exceeded 100 ng/mL. After handling AFM_1_ degradation products generated by rCAT, the percentage viability of Hep-G2 cells increased 2 times when compared with cells supplemented with an identical concentration of AFM_1_, indicating that the cytotoxicity of rCAT treated samples was less than AFM_1_.

Most toxicity analyses on the degradation products in the current study were theoretical speculations after unravelling toxic groups. Nikmaram et al. inferred that the construction of degradation products appeared to possess less toxicity than that of AFM_1_ because of the absence of a C8–C9 double bond and decoration of the furan ring [27]. Nguyen et al. reported the removal of the furan ring structure to form degradation products, which may have decreased the toxicity of these products compared to AFM_1_ [25]. Few studies have been carried out to experimentally analyze the toxicity of degradation products.

### 3.7. Molecular Docking and Binding Model of AFM_1_ and CAT

Docking simulation research was conducted to study the interplay between AFM_1_ and rCAT (Figure 6). The docking point for the bundling mode between AFM_1_ and rCAT was −7.3 kcal/mol. Hydrogen bonds and Van der Waal forces played major roles in the stability of AFM_1_–rCAT_1_. There was a hydrogen bond shaped between rCAT_1_ and AFM_1_, which was connected to the carbonyl group of the ligand. AFM_1_ had a Van der Waal interaction with Phe-227 in the receptor. These analyses were consistent with test results, indicating that AFM_1_ was a new substrate for CAT.

### 3.8. Degradation of AFM_1_ in Milk and Beer

The degradation values for AFM_1_ in beer and milk by rCAT were 31.3% and 47.2%, respectively (Table 2). Compared with the pattern solution, these lower degradation rates may have been related to acidity in the beer (pH 4.41) and milk (pH 6.25), resulting in variations in the active position of the enzyme or lack of ionization of aflatoxin [24]. In addition, the rCAT was a neutral enzyme coupled with that milk had a higher pH than beer, and rCAT was usually degraded AFM_1_ at a lower rate than milk. It was inferred that acid conditions would make AFM_1_ degradation less effective. Therefore, it is essential to modify a protein to make it more suitable for acid environments.

A previous study observed that rCAT from Chinese black sleeper (*Bostrychus sinensis*) participated in the oxidative homeostasis of the sleeper during pathogen intrusion [28]. Shaeer et al. found that a recombinant manganese-catalase from *Geobacillus thermopakistaniensis* had high activity and heat resistance, making it a promising candidate for industrial applications [18]. Thus, catalase is a highly valuable protein. However, few studies have been conducted to examine the degradation of AFM_1_ by rCAT. In the previous literature, a rCAT from *Pseudomonas aeruginosa* degraded AFB_1_ in the presence of syringaldehyde by 38.8% [16].

We investigated the degradation of AFM_1_ by rCAT from *B. pumilus*E-1-1-1 and found that rCAT degraded more than 60% of the toxin. In pattern solution, the optimal reaction conditions were pH 7, 40 °C, and the addition of K^+^. Degradation rates in lager beer and milk were 31.3% and 47.2%, respectively. Given food flavor directions, this study did not further improve the degradation situation in beer and milk, and recombinase embellishments could be considered to increase its degradation ability and enhance its tolerance to acid and alkali. Our purified rCAT effectively catalyzed the immediate oxidation of AFM_1_ without redox mediators. Based on LC–MS analyses, the rCAT-mediated AFM_1_ oxidation product was deduced as C_15_H_14_O_3_. Toxicological analyses revealed that C_15_H_14_O_3_ was lower than the same concentration of AFM_1_ itself. In short, the rCAT-mediated degradation of AFM_1_ in pattern solution, beer, and milk has shown that it could be used as an alternative product in the food industry, and lays the foundation for other research on AFM_1_ degradation in food materials.

## Figures and Tables

**Figure 1 foods-13-00888-f001:**
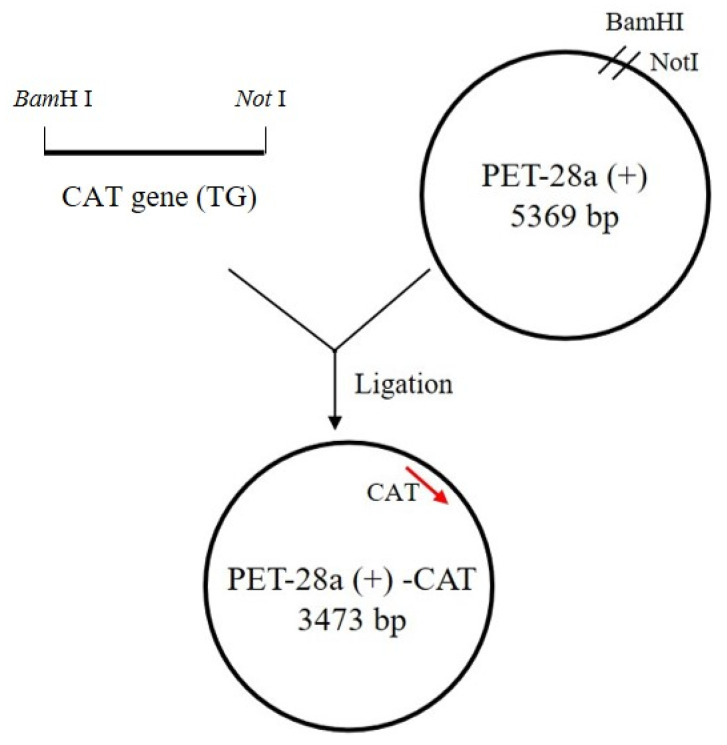
Physical maps of the recombinant expression vector PET28a–CAT. The red arrow indicates the target gene.

**Figure 2 foods-13-00888-f002:**
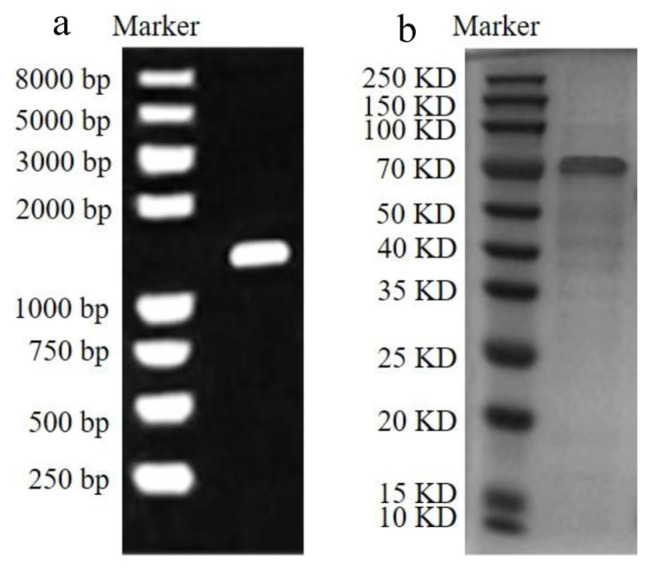
Agarose gel electrophoresis and SDS-PAGE of recombinant CAT. The left lane is the marker and the right lane is the target gene/protein. (**a**) PCR detection of recombinant CAT. (**b**) Solubility analysis and recombinant CAT purification on an Ni^2+^-NTA affinity column.

**Figure 3 foods-13-00888-f003:**
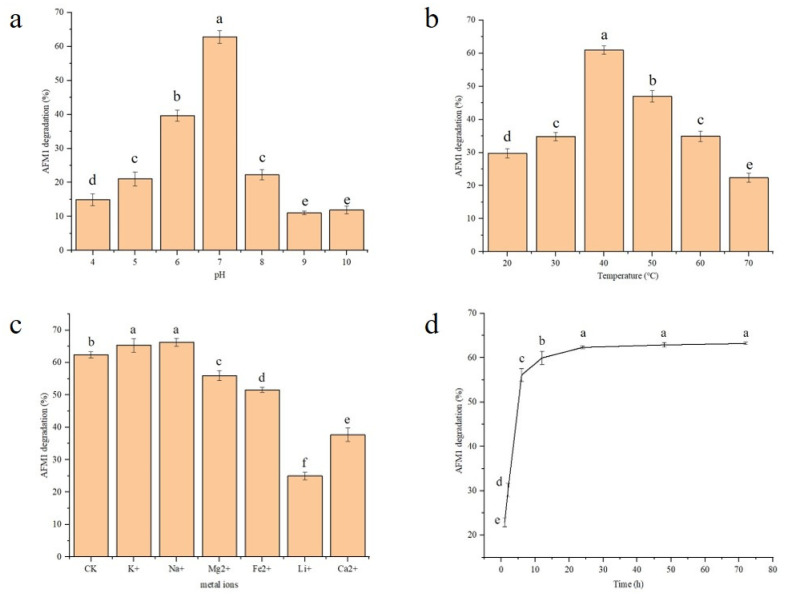
The effects of pattern solution pH, temperature, metal ions, and degradation kinetics on AFM_1_ degradation by rCAT. (**a**) A 24-h study of AFM_1_ degradation by rCAT under different pHs. Reaction conditions: pH 4–10, 30 °C, initial AFM_1_ concentration 200 ng/mL, and rCAT activity 1 U/mL. (**b**) The effect of temperature on AFM_1_ degradation. Reaction conditions: pH 9, 20–70 °C, initial AFM_1_ concentration 200 ng/mL, and rCAT activity 1 U/mL. (**c**) The effects of metallic ion addition on rCAT activity using pyrogallol autoxidation methods. Pattern solution—50 mM Tris-HCl at pH = 9, 1 mM metallic ions, 50 mM pyrogallol, and 1 U/mL rCAT; incubated at 40 °C for 24 h. CK-enzyme activity was recorded without adding metallic ions during AFM_1_ degradation. (**d**) The AFM_1_ degradation rates by rCAT within 72 h. a–f *p* < 0.05.

**Figure 4 foods-13-00888-f004:**
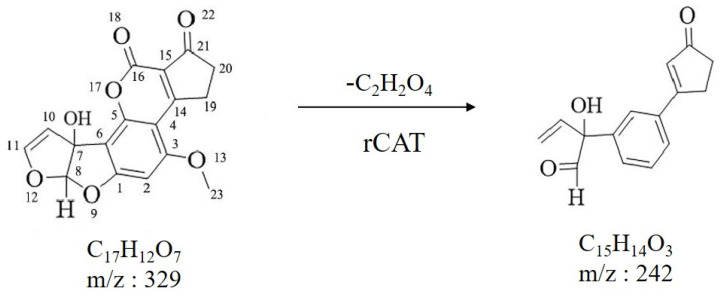
The reaction scheme for AFM_1_ oxidation by rCAT. The numbers are obtained by labelling the positions of the carbon atoms in the aflatoxin counterclockwise.

**Figure 5 foods-13-00888-f005:**
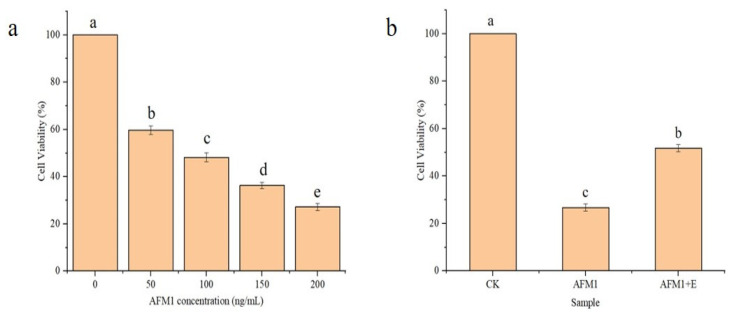
The effects of AFM_1_ and its catalase–generated degradation products on Hep-G2 cell viability. (**a**) The effects of AFM_1_ on Hep-G2 cell viability. The concentration of AFM_1_ was 0, 50, 100, 150, and 200 ng/mL, respectively. (**b**) The effects of catalase–generated degradation products on Hep-G2 cell viability. CK indicates the control (without AFM_1_); AFM_1_ indicates the 200 ng/mL AFM_1_ group; AFM_1_ + E indicates the catalase–generated degradation products of 200 ng/mL AFM_1_. Different letters indicate significant differences between levels evaluated for the same AFM_1_ (*p* < 0.05).

**Figure 6 foods-13-00888-f006:**
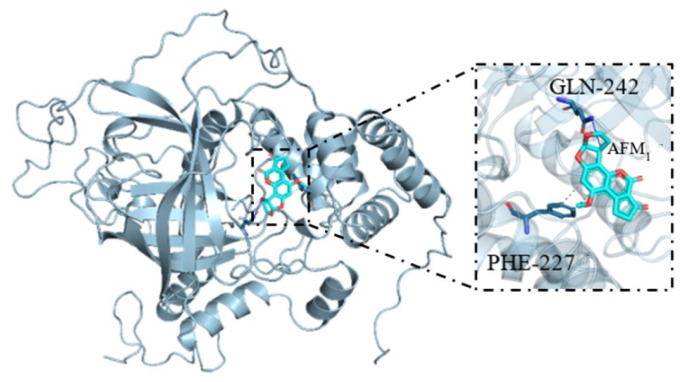
Binding model of AFM_1_ to CAT by molecular docking. The 3D interaction model of AFM_1_ with CAT. The right part is an enlarged view of the left part. AFM_1_ is in light blue, and the surrounding residues in binding pockets are in dark blue. The CAT backbone is depicted as a light gray ribbon.

**Table 1 foods-13-00888-t001:** Forward and reverse primers.

Name	Sequence	Length
Forward	CGGGATCCATGAAAGAAGATCAACACCCTAAG	32
Reverse	TTGCGGCCGCTTAATAAGGATCTGATGGTGTG	32

**Table 2 foods-13-00888-t002:** Degradation rates of AFM_1_ by recombinant catalase in beer and milk.

Name	Degradation
beer	31.3%
milk	47.2%

## Data Availability

Data sets produced and assayed in this research were obtained from corresponding authors according to rightful requirements.

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
