# Peer review of "Degradation Mechanism of Aflatoxin M1 by Recombinant Catalase from Bacillus pumilusE-1-1-1: Food Applications in Milk and Beer"

_foods, 2024, doi:10.3390/foods13060888_

Round 1

Reviewer 1 Report

Comments and Suggestions for Authors

--Manuscript ID- foods-2629855-- Review

Many points need to be reviewed.

General points

Point 1

I suggest changing the title with:

“Degradation mechanism of Aflatoxin M1 by recombinant catalase from Bacillus pumilusE-1-1-1: food applications in milk and beer” 

Point 2

Integrate abstract section with the following information:

1.     indicate precisely from which animal the Bacillus pumilus strain was isolated;

2.     indicate in extreme summary the technique of isolation and identification of the bacterial strain;

3.     highlight the eco-friendly advantages of the biological decontamination of aflatoxin compared to the risks of traditional physical and chemical treatments.

Punto 3

 The references section must be carefully reviewed: many references are not relevant (some are indicated in the following specific points).

Also review and standardize the punctuation according to the journal's standards.

Specific points

Materials and methods

2.1

B. pumilusE-1-1-1 was screened from animal feces and kept in the laboratory.

which animal??

indicate in detail methods of isolation and preservation of the strain.

Rewrite the point clearly!!

2.2

The PCR amplification product of expression vector PET28a(+)???

The CAT gene was obtained from the chromosome of B. pumilus? True? So, the correct version is:” the expression vector for cloning of PCR amplification product”.

The point should be rewritten better by also including the part of the methods concerning the extraction of bacterial genomic DNA used as atemplate for the amplification procedures.

Rewrite the point clearly!!

2.9

Use uppercase acronym for elisa

3.1.

In this point the activity of CAT on hydrogen peroxide was tested, so references 16 and 17 are not relevant.Replace!!

3.2.2

rCATs from the same genus (B. subtilis) had identical optimal reaction temperature of 40°C.

Bibliographical references are missing for this discussion point!! Why??

3.6

In figure 5b, also in the figure caption, the concentration of AFM1 and AFM +E used in ng/mL must be specified,

“Sample” should be omitted.

References

Reference [1] not relevant. Replace with Liu et al., 2011

Also reference [2] not relevant. In the paper indicated, no aflatoxins other than AFB1 are mentioned. Replace!!!

Check all references carefully.

Author Response

Title: Degradation mechanism and application in milk and beer of Aflatoxin M1 by recombinant catalase from Bacillus pumilusE-1-1-1

Journal: Foods

Article ID: Foods-2629855

Dear editor,

I'm glad that you put forward your valuable suggestions. We would like to express our gratitude for your efforts in reviewing our manuscript (Foods-2629855) and for the valuable comments and suggestions to improve the quality of the paper. We have studied the comments carefully and tried our best to revise the manuscript accordingly. All revised portions are highlighted in red color in the manuscript. The point-to-point responses to the reviewers’ comments are as follows (the corrected portions are highlighted in red text in manuscript):

Reviewer’s comment: I suggest changing the title with: “Degradation mechanism of Aflatoxin M1 by recombinant catalase from Bacillus pumilusE-1-1-1: food applications in milk and beer”

Response: Thank you for your suggestion. The title has been changed to the new one in manuscript with red words.

Reviewer’s comment: Integrate abstract section with the following information:

1.indicate precisely from which animal the Bacillus pumilus strain was isolated;

2.indicate in extreme summary the technique of isolation and identification of the bacterial strain;

3.highlight the eco-friendly advantages of the biological decontamination of aflatoxin compared to the risks of traditional physical and chemical treatments.

Response: Thank you for your suggestion. The above three suggestions have been added to the beginning of the abstract in the manuscript and highlighted in red (Line 27-32). Meanwhile, the relatively more detailed steps could be found at the beginning of 2.1 and were highlighted in red (Line 98-101).

Reviewer’s comment: The references section must be carefully reviewed: many references are not relevant (some are indicated in the following specific points).

Also review and standardize the punctuation according to the journal's standards.

Response: Thank you for your suggestion. The references section has been reviewed and corrected, and those that can be labelled have been screenshotted at the bottom of the document. The list of authors and co-authors have been completed in the manuscript and marked in red.

Reviewer’s comment: Materials and methods

2.1 B. pumilusE-1-1-1 was screened from animal feces and kept in the laboratory.

which animal?? Indicate in detail methods of isolation and preservation of the strain.

Rewrite the point clearly!!

Response: Thank you for your suggestion. Specific steps for strain screening and preservation have been added to the beginning of 2.1 and were marked in red (Line 98-101).

Reviewer’s comment: 2.2 The PCR amplification product of expression vector PET28a(+)??? The CAT gene was obtained from the chromosome of B. pumilus? True? So, the correct version is:” the expression vector for cloning of PCR amplification product”.

The point should be rewritten better by also including the part of the methods concerning the extraction of bacterial genomic DNA used as a template for the amplification procedures.

Rewrite the point clearly!!

Response: Thank you for your suggestion. This part of the expression has been supplemented and modified in the manuscript and was marked in red (Line 105-107 and Line 112).

Reviewer’s comment: 2.9 Use uppercase acronym for elisa

Response: Thank you for your suggestion. The word “elisa” has been changed to word “ELISA” in manuscript and was marked in red (Line 196).

Reviewer’s comment: 3.1. In this point the activity of CAT on hydrogen peroxide was tested, so references 16 and 17 are not relevant. Replace!!

Response: Thank you for your suggestion. The two previous references have been replaced by two new articles on recombinant catalase enzyme activity. Two deleted literatures were marked in light grey in the original text and two newly added literatures were marked in red.

Reviewer’s comment: 3.2.2 rCATs from the same genus (B. subtilis) had identical optimal reaction temperature of 40°C. Bibliographical references are missing for this discussion point!! Why??

Response: Thank you for your suggestion. The reference to this sentence has been added to the end of the sentence (Line 265).

Reviewer’s comment: 3.6 In figure 5b, also in the figure caption, the concentration of AFM1 and AFM1 +E used in ng/mL must be specified, “Sample” should be omitted.

Response: Thank you for your suggestion. The concentration of AFM1 and AFM1 + E has been added in figure 5b and was marked in red (Line 555 and Line 556). The word “sample” has been omitted.

Reviewer’s comment: References

Reference [1] not relevant. Replace with Liu et al., 2011

Also reference [2] not relevant. In the paper indicated, no aflatoxins other than AFB1 are mentioned. Replace!!! Check all references carefully.

Response: Thank you for your suggestion. The references section has been reviewed and corrected, and those that can be labelled have been screenshotted at the bottom of the document. The list of authors and co-authors have been completed in the manuscript and marked in red.

[3]

[4]

[5]

[6]

[7]

[8]

Reviewer 2 Report

Comments and Suggestions for Authors

The authors should consider the following suggestions:

Abstract:

Change the sentence as follow

This work aimed at studying the effect of a recombinant cat…

Throughout all the manuscript, please check carefully all percents and round with one decimal place (ex: 31.3% and 47.2% instead of 31.27% and 47.21%).

1. Introduction:

Aflatoxin is a naturally occurring fungal toxic metabolite, primarily generated by Aspergillus flavus and Aspergillus parasiticus, with high carcinogenicity and muta-genicity [1]. There are 18 types of aflatoxins including aflatoxin B1 (AFB1), aflatoxin B2 (AFB2), afla-toxin G1 (AFG1), aflatoxin G2 (AFG2), aflatoxin M1 (AFM1) and aflatoxin M2 (AFM2) [2]. AFB1 is the most toxic and carcinogenic. AFM1 is a hydroxyl metabolite of AFB1 with similar poisonousness and often exist in milk and, subsequently, in other dairy products [3]. The study of AFM1 in food is of significance as may pose a special risk to children. In recent years, the breweries have been developed, and beer consumption in China has increased.

2.12. Statistical analysis

Please correct for SPSS  Statistical Package for Social Sciences in the sentence “Statistical Product and Service Solution (SPSS 22.0, USA) was used to analyze….”

3.4 Determination of kinetic parameters

Please specify the document…”In the document ???….,

3.5. identification of degraded products by LC-MS

Capitalize the initial

References:

Complete the list of authors and co-authors for the following citations…

1.     Guo. Z. et al. (2023).

2.     Zhu. H. F. et al. (2023).

3.     Giacometti. F. et al. (2023).

4.     Jiang. M. M. et al. (2023).

5.     Torshizi. M. A. K. et al. (2023).

6.     Guo. Y. P. et al. (2021).

7.     Sheen. A. et al. (2023).

8.     Qi, L.G. et al. (2023).

9.             Li. W. et al. (2023).

 etc…

Comments on the Quality of English Language

Dear

Moderate editing of english improvement is needed...

Best regards

Author Response

Title: Degradation mechanism and application in milk and beer of Aflatoxin M1 by recombinant catalase from Bacillus pumilusE-1-1-1

Journal: Foods

Article ID: Foods-2629855

Dear editor,

I'm glad that you put forward your valuable suggestions. We would like to express our gratitude for your efforts in reviewing our manuscript (Foods-2629855) and for the valuable comments and suggestions to improve the quality of the paper. We have studied the comments carefully and tried our best to revise the manuscript accordingly. All revised portions are highlighted in red color in the manuscript. The point-to-point responses to the reviewers’ comments are as follows (the corrected portions are highlighted in red text in manuscript):

Reviewer’s comment: Abstract: Change the sentence as follow: This work aimed at studying the effect of a recombinant CAT… Throughout all the manuscript, please check carefully all percents and round with one decimal place (ex: 31.3% and 47.2% instead of 31.27% and 47.21%).

Response: Thank you for your suggestion. The sentence has been changed to “this work aimed at studying the effect of a recombinant CAT (Line 31-32) … All percents have been changed to round with one decimal place throughout the manuscript.

Reviewer’s comment: 1. Introduction: Aflatoxin is a naturally occurring fungal toxic metabolite, primarily generated by Aspergillus flavus and spergillus parasiticus, with high carcinogenicity and mutagenicity [1]. There are 18 types of aflatoxins including aflatoxin B1 (AFB1), aflatoxin B2 (AFB2), aflatoxin G1 (AFG1), aflatoxin G2 (AFG2), aflatoxin M1 (AFM1) and aflatoxin M2 (AFM2) [2]. AFB1 is the most toxic and carcinogenic. AFM1 is a hydroxyl metabolite of AFB1 with similar poisonousness and often exist in milk and, subsequently, in other dairy products [3]. The study of AFM1 in food is of significance as may pose a special risk to children. In recent years, the breweries have been developed, and beer consumption in China has increased.

Response: Thank you for your suggestion. Thank you for your patience in correcting this section, which has been fixed and highlighted in red (Line 52-58).

Reviewer’s comment: 2.12. Statistical analysis; Please correct for SPSS Statistical Package for Social Sciences in the sentence “Statistical Product and Service Solution (SPSS 22.0, USA) was used to analyze….”

Response: Thank you for your suggestion. The word “Statistical Product and Service Solution” has been changed to “Statistical Package for Social Sciences” and marked in red (Line 211).

Reviewer’s comment: 3.4 Determination of kinetic parameters

Please specify the document…” In the document ???….,

Response: Thank you for your suggestion. The document has been changed to literature and marked in red (Line 288 and Line 364).

Reviewer’s comment: 3.5. identification of degraded products by LC-MS

Capitalize the initial

Response: Thank you for your suggestion. The word “identification” has been changed to word “Identification” (Line 294).

Reviewer’s comment: References: Complete the list of authors and co-authors for the following citations…

1.Guo. Z. et al. (2023). 2.Zhu. H. F. et al. (2023). 3.Giacometti. F. et al. (2023). 4.Jiang. M. M. et al. (2023). 5.Torshizi. M. A. K. et al. (2023). 6.Guo. Y. P. et al. (2021). 7.Sheen. A. et al. (2023). 8.Qi, L.G. et al. (2023). 9.Li. W. et al. (2023). Etc.

Response: Thank you for your suggestion. The list of authors and co-authors have been completed in the manuscript and marked in red.

Reviewer 3 Report

Comments and Suggestions for Authors

The work presented for review, although touching on a rather interesting topic, needs to be improved.

Especially due to:

Lack of culture conditions, methodology for obtaining catalase solution.

Why was this particular strain of bacteria chosen?

It is required to clearly emphasise the validity of the research undertaken and to indicate its impact.

Author Response

Title: Degradation mechanism and application in milk and beer of Aflatoxin M1 by recombinant catalase from Bacillus pumilusE-1-1-1

Journal: Foods

Article ID: Foods-2629855

Dear editor,

I'm glad that you put forward your valuable suggestions. We would like to express our gratitude for your efforts in reviewing our manuscript (Foods-2629855) and for the valuable comments and suggestions to improve the quality of the paper. We have studied the comments carefully and tried our best to revise the manuscript accordingly. All revised portions are highlighted in red color in the manuscript. The point-to-point responses to the reviewers’ comments are as follows (the corrected portions are highlighted in red text in manuscript):

Reviewer’s comment: The work presented for review, although touching on a rather interesting topic, needs to be improved. Especially due to: Lack of culture conditions, methodology for obtaining catalase solution.

Response: Thank you for your suggestion. The culture conditions and the methodology for obtaining catalase solution have been marked in red in the manuscript (Line 124-126 and Line 131).

Reviewer’s comment: Why was this particular strain of bacteria chosen?

Response: Thank you for your suggestion. This strain was screened from Africa elephant feces for its AFM1 degrading ability and was of high research value. The screening, identification and preservation process have been added to the abstract and the beginning of 2.1 (Line 27-29 and Line 101-104).

Reviewer’s comment: It is required to clearly emphasis the validity of the research undertaken and to indicate its impact.

Response: Thank you for your suggestion. The impact of this research has been marked in red (Line 376-379).

Round 2

Reviewer 3 Report

Comments and Suggestions for Authors

The revisions have improved the publication and I wish the authors success in their continued research.

Author Response

Title: Degradation mechanism and application in milk and beer of Aflatoxin M1 by recombinant catalase from Bacillus pumilusE-1-1-1

Journal: Foods

Article ID: Foods-2629855

Dear editor,

I'm glad that you put forward your valuable suggestions. We would like to express our gratitude for your efforts in reviewing our manuscript (Foods-2629855) and for the valuable comments and suggestions to improve the quality of the paper. We have studied the comments carefully and tried our best to revise the manuscript accordingly. All revised portions are highlighted in red color in the manuscript. The point-to-point responses to the reviewers’ comments are as follows (the corrected portions are highlighted in red text in manuscript):

Reviewer’s comment: Abstract: Spaces missing between pumilus and de code E-1-1-1 and numbers and microgram units. Spaces between numbers and units should be checked through all the manuscript.

Response: Thank you for your suggestion. Spaces between figures and units have been checked and corrected throughout the whole manuscript and highlighted in red.

Reviewer’s comment: Introduction: In this sentence there is a verb missing, please revise: it only be controlled within a relatively safe dosage range…

Response: Thank you for your suggestion. The missing verb has been added in the manuscript and marked in red (Line 60).

Reviewer’s comment: Material and methods: Authors should explain from where they obtained the HepG2 cells and which range of passages are they using for the experiments.

Response: Thank you for your suggestion. The sources and periodicity of the HepG2 cells were added in the manuscript and marked in red (Line 182-183).

Reviewer’s comment: In this sentence I think they mean add instead of increased: “CCK-8 reagent solution was increased to each well”

Response: Thank you for your suggestion. The word “increased” has been changed to the word “added” and marked in red (Line 191).

Reviewer’s comment: Change “absorbency” for absorbance, as it is related to a wavelength.

Response: Thank you for your suggestion. The word “absorbency” has been changed to the word “absorbance” and marked in red (Line 193).

Reviewer’s comment: In section 2.11 there is a lot of information missing, like AFM1 concentration in beer and milk, the volume used for diluting again the dried amount of mycotoxin in beer and milk, the temperature and time applied to the rCAT to degradate AFM1, sample preparation for AFM1 quantification by HPLC-MS-TOF and method for quantification.

Response: Thank you for your suggestion. The missing information has been added in the manuscript and marked in red (Line 206-207 and Line 210).

Reviewer’s comment: Results: In section 3.6 authors write density when they mean concentration of AFM1. In section 3.8, the most important results of this manuscript are the less explained, not even a figure, which would be very much appreciated.

Response: Thank you for your suggestion. The word “density” has been changed to the word “concentration” and marked in red in section 3.6. In section 3.8, the most important results of this manuscript have been explained in Table 2 at the last of the manuscript and marked in red.
